# Intercropping in Rice Farming under the System of Rice Intensification—An Agroecological Strategy for Weed Control, Better Yield, Increased Returns, and Social–Ecological Sustainability

**Tavseef Mairaj Shah** [1,*] , **Sumbal Tasawwar** [1] , **M. Anwar Bhat** [2] **and Ralf Otterpohl** [1]

1    Rural Revival and Restoration Engineering, Institute of Wastewater Management and Water Protection, Hamburg University of Technology, 21073 Hamburg, Germany; sumbal.tasawwar@tuhh.de (S.T.); ro@tuhh.de (R.O.)
2    Division of Agronomy, Faculty of Agriculture, Sher-e-Kashmir University of Agricultural Sciences and Technology of Kashmir, Sopore 193201, Jammu & Kashmir, India; hodagron_foa@skuastkashmir.ac.in
*    Correspondence: tavseef.mairaj.shah@tuhh.de

**Abstract:** Rice is the staple food for more than half of the world's population. In South Asia, rice farming systems provide food to the majority of the population, and agriculture is a primary source of livelihood. With the demand for nutritious food increasing, introducing innovative strategies in farming systems is imperative. In this regard, intensification of rice farming is intricately linked with the challenges of water scarcity, soil degradation, and the vagaries of climate change. Agroecological farming systems like the System of Rice Intensification (SRI) have been proposed as water-saving and sustainable ways of food production. This study examines the effect of intercropping beans with rice under SRI management on the growth of weeds and on the different plant growth parameters. Intercropping led to a 65% decrease in weed infestation on average, which is important given that weed infestation is stated as a criticism of SRI in some circles and is a major factor in limiting yield in rice-producing regions. In addition to the water savings of about 40% due to the SRI methodology, the innovation led to an increase in rice yield by 33% and an increase in the net income of farmers by 57% compared to the conventional rice farming method. The results indicate that intercropping can be a positive addition to the rice farming system, hence contributing to social–ecological sustainability.

**Keywords:** agroecology; rice; intercropping; sustainable agriculture; sustainable intensification

## 1. Introduction

Agriculture is at the core of one of the greatest challenges of this century: the challenges of meeting the neglected and the growing nutritional needs of the world's population and remedying the environmental damage due to agricultural activities at the same time [1]. Currently, 800 million individuals worldwide do not have access to enough food, while more than two billion people experience key micronutrient deficiencies. This is a problem that is more serious in low-income countries where the percentage of food insecure individuals is around 60% [2]. On the other hand, agriculture is now counted as a major force contributing to planetary overshoot of natural resources, contributing dominantly towards climate change, biodiversity loss, and the degradation of land and freshwater [3–5].

In this regard, rice farming systems are particularly relevant, with their contribution to feeding the world as well as the large scope for improvements with respect to different environmental issues associated with rice farming systems, specifically flooded rice cultivation systems [6–11]. Rice is a staple food for more than half of the world's population and contributes about one-fourth of the global energy consumption [12]. The wide-ranging importance of rice is evident from the fact that it is grown in 112 different countries of the

world in different climatic zones [12]. Irrigated rice cultivation covers about 60% of the total land under rice cultivation worldwide and contributes 75% to the total rice produce [13].

With a large water footprint, which is 2–3 times more than that of upland crops, irrigated flooded rice monocropping systems, dependent on agrochemical use, have been found to be detrimental to the environment and biodiversity [8]. For example, it has been reported that nitrogen losses due to ammonia volatilization account for up to 60% of nitrogen applied in flooded rice systems [8]. Flooding of rice paddies has also been found to have an association with health risks, particularly in tropical and sub-tropical regions, in terms of potentially aiding the proliferation of water-borne diseases [10]. Another health risk associated with flooded rice systems is the incidence of toxic residues of pesticides and metalloids/metals like arsenic, cadmium, and mercury in rice grains. This is particularly damaging for regions where rice forms a large percentage of the diet, for example, in South Asia and Southeast Asia. Arsenic pollution is a widespread problem in the Punjab and Bengal regions in South Asia, which straddle Pakistan, Bangladesh, and India [14–17]. In this regard, an aerobic water management system in rice, through the practice of alternate wetting and drying, has been reported to reduce the risk of uptake of metals like arsenic [18].

In this context, the increasing demand of nutritious food is putting an unprecedented pressure on the land and water resources. In this regard, 'another green revolution', which follows the methodology of the 'original' green revolution of the 1960s, is not a viable option, owing to its ecological and social costs [19]. Its positive effect on cereal production notwithstanding, the green revolution also contributed to the reduced availability of pulses and legumes (protein sources) in the rural diet [20]. Additionally, the per capita availability of land has halved from 1960 to 2010 and is expected to decrease by a further 30% by 2050 [21,22]. With respect to the livelihoods of farmers, stricter regulations on agrochemical residues in food grains are making agrochemical-based intensification unviable. It is in the backdrop of these socioeconomic challenges, highlighted and intensified during the COVID-19 pandemic, that sustainable intensification of agriculture is being proposed [23].

Agroecology-based farming strategies promise nutritious yields, with more consideration for the fast depleting natural resources vital for agriculture, i.e., water and soil. In a report by the FAO released in September 2020, the role of agroecology has been termed as vital for climate-resilient livelihoods and sustainable food systems [24]. The system of rice intensification (SRI) is an agroecological methodology of growing rice. It involves early plant establishment, wider spacing between plants transplanted individually, manual weeding, and alternate wetting and drying of the rice field instead of continuous flooding [25]. SRI-based rice farming systems have been associated with a range of environmental benefits, reportedly reducing water and energy use by 60% and 74%, respectively, and greenhouse gas emissions by 40% [11]. It has led to improved yields in different parts of the world and includes practices that enhance the soil health [26–29]. An oft-cited disincentive of this methodology, however, is the increased incidence of weeds due to the absence of continuous flooding of the fields, which creates aerobic conditions feasible for the growth of weeds [20,30,31]. This also leads to increased work-hour requirements for the weeding process [32]. Weeds are one of the single largest source of yield losses in rice farming, representing 6.6% of the yield gap in South Asia [33]. In the Mediterranean region as well, the occurrence of weeds has been described as the main reason behind yield variability and yield gap, specifically in organic rice farming systems [34–36]. Under SRI, without proper weed management, a reduction in rice yield by up to 70% has been reported due to weed infestation [37].

Hence, the incidence of weeds is reported as a challenge in SRI, along with the need for increased work-hours required during weeding, which occurs multiple times in a crop season [27,38]. In this regard, the availability of family labor has been found to influence the adoption of SRI practices by farm households [39]. Higher work-hour requirements may also have led to dis-adoption of SRI in some cases [40]. Intercropping has been reported as a natural way to control the growth of weeds in other cropping systems, including upland rice and maize farming systems [41–44]. The system of rice

intensification is relevant in this regard, as it provides wider inter-row space to introduce intercropping [45]. In addition to comparing the effect of intercropping on weed infestation, the effect of intercropping on different plant growth characteristics under SRI management has also been studied. Improvements in different plant growth characteristics under SRI management in comparison to the conventional flooded rice (CFR) has already been widely documented in literature [26,46–50].

The results of this study recommend the same in the case of rice grown under alternate wetting and drying conditions of SRI. It was hypothesized that growing a leguminous crop, i.e., beans, as an intercrop in between rice rows grown under SRI would affect the incidence of weeds and the growth characteristics of rice plants. These parameters were recorded in field experiments. The effect of intercropping on the chlorophyll content of the rice crop was also determined in multiple experiments under controlled lab greenhouse conditions. Growing beans together with rice can also serve to diversify the diet of the rural population by restoring an important source of protein to it. A new terminology, denoted as SRIBI, has been used for the methodology of system of rice intensification with beans intercropping. The conventional flooded rice method is represented as CFR.

## 2. Materials and Methods

### 2.1. Experimental Research Design

Experiments were first conducted in pots at the laboratory level in a greenhouse chamber under controlled conditions in three batches in the years 2017 and 2018. The pot experiments followed a mirrored randomized complete block design to avoid any variations due to placement in the greenhouse. Sandy clay loam soil was used in the pot experiments, which contained 11.25% organic matter, 0.16% total nitrogen, 0.05% total phosphorous, and 0.47% total potassium. In the first batch of experiments, the conventional flooded rice cultivation (CFR) was compared with the system of rice intensification (SRI) and SRI with beans intercropping (SRIBI). The CFR treatment is referred to as 'flooded' treatment, while the SRIBI treatment is referred to SRI+I in the case of pot experiments. Pots of diameter 25 cm and height 25 cm were used in this study. In these experiments, 4 replications of each treatment were analysed. Once it was established that intercropping had a positive effect on the rice crop under SRI management, the next experiments were conducted to find the ideal time and space combination between the rice and the intercrop (beans). Hence, the subsequent batches involved treatments I9 (intercropping done between rice rows at 9 days after transplantation), I35 (intercropping done between rice rows at 35 days after transplantation), and IS (intercropping done at 9 days after transplantation as strip intercropping), in addition to the standard SRI treatment. In these experiments, 8 replications of each treatment were analysed. These experiments were conducted in mini-plots of size 60 cm (length) by 50 cm (width) by 25 cm (height). Based on the analysis of plant growth characteristics like nutrient uptake, chlorophyll content, and yield parameters, the proof of concept was established and field experiments were designed accordingly, which were conducted in 2019 and 2020 (in the local rice growing season May–October).

The field experiments followed a randomized complete block design with four replications and a plot size of 60 m$^2$ (10 m × 6 m) each. The experiments were carried out in two villages falling under the Sagam belt of the Islamabad region (District Anantnag) in Kashmir, a popular niche belt of the local heritage aromatic landraces of Kashmir, particularly the Mushkibudij (Mushk Budji, literally 'Aromatic Grain') variety that was also used in the studies. The experimental fields were located at 33°36′31″ N, 75°14′59″ E and 33°36′54″ N, 75°15′2″ E, in Jammu and Kashmir. The soil at the experimental site was characterised as silty clay loamy soil, with a neutral pH of 7.3. The soil was low in available nitrogen (140–280 Kg/ha) and medium in available phosphorous (11–22 Kg/ha) and potassium (110–280 Kg/ha). The elevation of the experimental site is at 1800 m amsl. The average maximum temperatures over the months (May–October) were 25.98 °C and 27.14 °C in 2019 and 2020, respectively, while the average minimum temperatures for the same time

period were 12.01 °C and 12.19 °C, respectively. Total rainfall recorded in this period was 541.5 mm and 424.2 mm for 2019 and 2020, respectively.

For the CFR treatment, rice plants were transplanted according to the local conventions, at 5 weeks age. For field experiments, the control SRI treatment, in which weeds were allowed to grow for comparison, was referred to as SRI-w (Weedy control) to avoid any misunderstanding of the SRI method where weeding forms an integral part of the cultivation process. In the SRI-w and SRIBI treatments, rice plants were uniformly transplanted in a square pattern at a distance of 25 cm from each other, following the SRI method. Following the SRI method, seedlings were transplanted singly at the two-leaf stage (10 days after sowing). For the CFR treatment, NPK fertilizer was applied based on the standard application: 300 kg (150 N + 75 $P_2O_5$ + 75 $K_2O$) per ha. For the SRI and SRIBI treatments, only compost was applied at the rate of 6 t per ha. No pesticides were used in the SRI and SRIBI treatments. The details of individual practices followed in the three treatments are given in Table 1.

**Table 1.** A comparison of the different agricultural practices in standard SRI (System of Rice Intensification) and the two other SRI-based treatments in the current study.

| Practice\System | SRI | SRI-w (Weedy Control) | SRIBI |
|---|---|---|---|
| *Early Transplanting* | Yes | Yes | Yes |
| *Wide Spacing* | Yes | Yes | Yes |
| *One Plant per Hill* | Yes | Yes | Yes |
| *Compost Application* | Yes | Yes | Yes |
| *Alternate Wetting and Drying* | Yes | Yes | Yes |
| *Frequent Weeding* | Yes | No | No |
| *Intercropping* | No | No | Yes |

The SRI plots in this study were not weeded after the first weeding and were used as the weedy-control (SRI-w). Hence, this study is not an evaluation of the yield potential of SRI per se, but rather a study to determine how far intercropping can contribute to make the SRI method better.

### 2.2. Data Collection

The data on the reported parameters of weed density, plant height, tiller number, panicle length, and spikelet number per panicle were collected manually after 120 days after transplantation, which corresponds to the ripening stage of the rice crop. Work-hours were recorded during the course of the experiments. The occurrence of rice blast was recorded on the basis of on-farm experiences.

### 2.3. Laboratory Analysis

The chlorophyll content was measured following the method described by Arnon [51] used by Doni et al. [49,52]. The chlorophyll content in leaves was measured at different growth stages, i.e., at the seedling stage and then at different days after transplantation, using the following formulae:

$$C_{Chl-a} = 12.7\ A_{663} - 2.69\ A_{645};\ C_{Chl-b} = 22.9\ A_{645} - 4.68\ A_{663}$$

where $A_{663}$ and $A_{645}$ are the values of absorbance of the solution at wavelengths of 663 nm and 645 nm, respectively. The solution was prepared by cutting leaves into fine pieces and placing 0.1 g of the same pieces into a test-tube to which 20 mL of 80% acetone was added. This solution was kept in the dark for 48 h for incubation, and afterwards, it was analysed with a spectrophotometer.

Nutrient uptake in the plants was measured at the maturity stage (120 days). Rice plants were washed with water after harvesting and were allowed to dry, covered in paper bags at 65 °C for 7 days. A mixture of all plant parts was then ground up to pass through a 1 mm sieve. This ground mixture was then analysed for NPK content. Nitrogen was measured by the Kjeldahl method, phosphorus by cuvette test (Hach Lange LCK350), and potassium by the reflectometer method (Merck Reflectoquant RQflex 10 Reflectometer). Nitrogen content was also measured by a cuvette test (Hach Lange LCK138) and was found to be the same as the value determined using the Kjeldahl method.

### 2.4. Statistical Analysis

The statistical analysis for ANOVA (Analysis of Variance) was performed using Microsoft Excel 2013 (Microsoft Corporation, Redmond, DC, USA, https://www.microsoft.com/ Last accessed on: 19 May 2021). The statistical significance level in the ANOVA was set at $p \leq 0.05$. Post hoc analysis of ANOVA was also done using the Tukey–Kramer test for parameters involving comparisons of three treatments.

## 3. Results and Discussion

The experiments were carried out over a period of four years (2017–2020), with pot experiments in 2017–2018 and field experiments in 2019 and 2020. Certain parameters, like nutrient uptake, chlorophyll content, and tiller number, were measured in the lab experiments (pot experiments) to establish the proof of concept about any positive effects intercropping has on the growth of rice under the system of rice intensification (SRI). These parameters were measured for conventional flooded rice (CFR), SRI, and SRI with intercropping (SRIBI) in 2017, while in 2018, different intercropping configurations were compared with SRI. These configurations included I9/SRIBI-9 (intercropping at 9 days after transplantation, DAT), I35/SRIBI-35 (intercropping at 35 DAT), and IS (strip intercropping at 9 DAT). In the field experiments, the control SRI treatment was referred to as SRI-w (weedy control), which was compared to the intercropping treatment (SRIBI). In the pot experiments, the intercropping treatment (SRIBI) was referred to as SRI+I, while CFR was referred to as 'flooded' treatment. The detailed statistical data about the different measured parameters is included in Appendix A.

### 3.1. Field Experiments
#### 3.1.1. Weed Incidence

The growth of weeds was compared in the treatments of SRI-w (weedy-control) and SRIBI (system of rice intensification with beans intercropping). The number of weed species found in the two treatments was the same, indicating similar growth conditions for the weeds. In total, six weed species were found to be present: Echinochloa colona, Cynodon dactylon, Ammania baccifera, Cyperus iria, Cyperus deformis, and Fimbristylis. However, the density of weeds was observed to be significantly less in the case of the plots with intercropping ($p \leq 0.05$; $p = 0.0048$) (Table 2). The mean values of weed density for plots with and without intercropping were 56 per m$^2$ and 196 per m$^2$, respectively, in 2019, as can be seen from Figure 1. In 2020, the mean values of weed density of plots with and without intercropping were 87 and 213, respectively, as shown in Figure 1. On an average, intercropping led to a decrease in weed incidence by 65% in the field experiments over two years. A reduction in the weed density under intercropping regimes has previously been observed in upland rice and dry seeded rice cropping systems [43]. Intercropping of Sesbania in between the crop rows has been found to increase soil fertility in addition to suppressing the growth of weeds [41]. Weeds have also been found to be more responsive to the nitrogen applied to the soil, as a result of which more yield losses and lower values of crop growth parameters are expected in the presence of weeds [42]. The following parameters measured in the current study exemplify this effect.

**Table 2.** Parameters from ANOVA statistical analysis for weed density comparison.

| *Weed Density* | 2019 SRI-w vs. SRIBI | | | | 2020 SRI-w vs. SRIBI | | | |
|---|---|---|---|---|---|---|---|---|
| | *p*-value | F-value | $F_{critical}$ | DF | *p*-value | F-value | $F_{critical}$ | DF |
| | 0.0048 | 14.96 | 5.32 | (1,8) | 0.0001 | 51.81 | 4.74 | (1,12) |

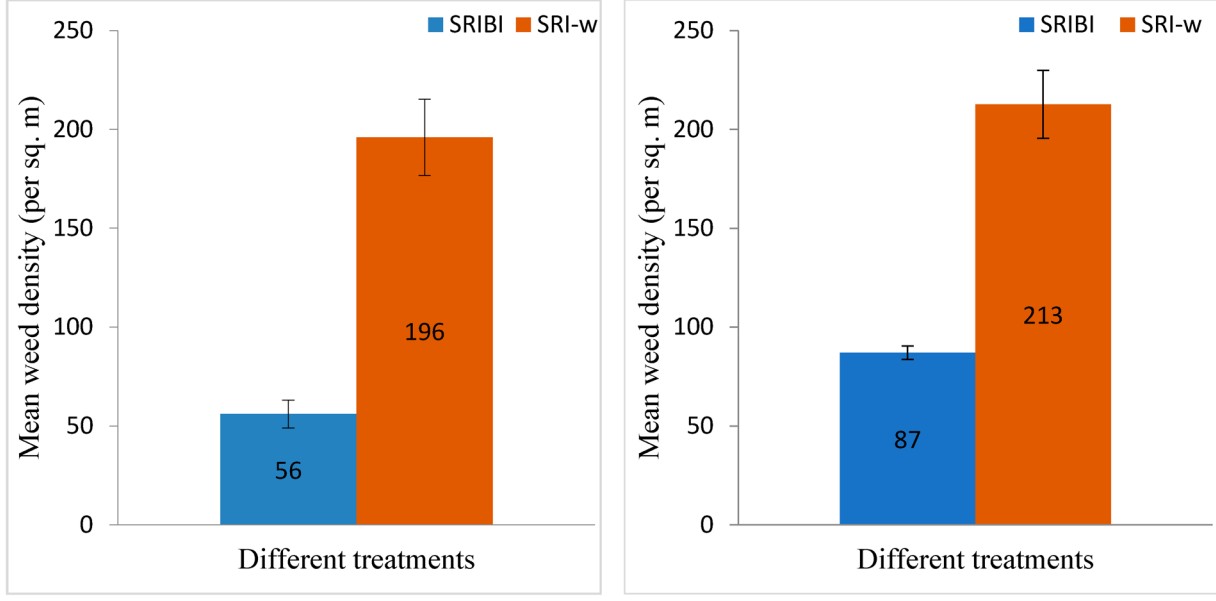

**Figure 1.** Weed density in the two SRI-based treatments (SRI-w: SRI weedy control; SRIBI: SRI with beans intercropping) (**Left**: 2019; **Right**: 2020).

The decreased weed incidence under the intercropping regime, as seen in Figure 2, makes the adoption of SRI as an agroecological methodology easier for the farmers. Weed infestation is otherwise a main criticism of the SRI methodology and has led to dis-adoption in some regions. The decreased weed incidence can also decrease the dependence of smallholder farmers on chemical solutions to tackle weeds. This can, in turn, lead to less contamination of soil and water which results from excessive use of agrochemicals. The lower weed incidence also leads to a lower labour requirement, which provides another incentive to those farmers for whom SRI is otherwise labour-intensive.

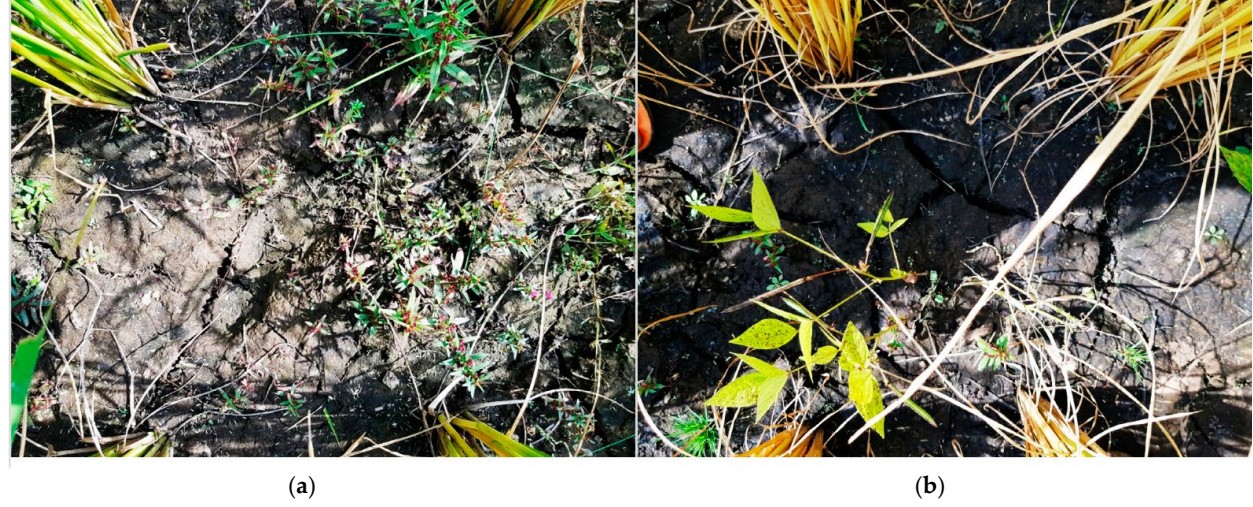

**(a)**            **(b)**

**Figure 2.** A comparison of weed infestation: (**a**) SRI weedy control, SRI-w; (**b**) SRI with intercropping, SRIBI (2019).

### 3.1.2. Plant Growth Parameters

Plant Height

A comparison of the plant height between rice with and without intercropping suggests that the lower weed density had visible effects on the growth of rice under the two dry rice systems. Rice plants under the intercropping regimen (SRIBI) showed higher plant height as compared to those without intercropping (SRI-w) (Tables 3 and 4. The mean values for plant height were 127.50 cm, 119.25 cm, and 125 cm for SRIBI, SRI-w, and CFR, respectively, at 120 days after transplantation, in 2019 (Table 3). In 2020, the observed mean values for plant height for CFR, SRI-w, and SRIBI were 101 cm, 104 cm, and 109 cm, respectively (Table 4). The effect on the plant height due to intercropping was observed to be significant ($p \leq 0.05$; $p = 0.05$) (Table 5). The maximum and mean heights observed in the three treatments are shown in Tables 3 and 4. The better plant height and earlier harvest-readiness in the intercropping treatment can be attributed to nitrogen fixation by the intercropped legumes as well as the higher availability of applied nitrogen to the rice crop with lower weed density [41,42].

**Table 3.** A comparison of plant height parameters observed for the three different treatments (2019).

| Index | Crop Management System | | |
|---|---|---|---|
| | **CFR** | **SRI-w** | **SRIBI** |
| Maximum height (cm) | 136 | 125 | 148 |
| Mean height (cm) | 125 | 119 | 128 |

**Table 4.** A comparison of plant height parameters observed for the three different treatments (2020).

| Index | Crop Management System | | |
|---|---|---|---|
| | **CFR** | **SRI-w** | **SRIBI** |
| Maximum height (cm) | 102 | 106 | 112 |
| Mean height (cm) | 101 | 104 | 109 |

**Table 5.** Parameters from ANOVA statistical analysis for plant height comparison.

| Plant Height | 2019 SRI-w vs. SRIBI | | | | 2020 CFR vs. SRI-w vs. SRIBI | | | |
|---|---|---|---|---|---|---|---|---|
| | *p*-value | F-value | $F_{critical}$ | DF | *p*-value | F-value | $F_{critical}$ | DF |
| | 0.05 | 4.50 | 4.49 | (1,14) | 0.0001 | 35.73 | 3.88 | (2,12) |
| | | | | | Post hoc analysis | | | |
| | | | | | $Q_{critical}$ | $AMD_{CFR/SRI}$ | $AMD_{SRI/SRIBI}$ | $AMD_{CFR/SRIBI}$ |
| | | | | | 4.72 | 3 | 5.8 | 8.8 |

Post hoc tests were also done for plant growth parameters measured in the field experiments, using the Tukey–Kramer test. The absolute mean difference is used to test the significance of the difference of means. In case of the observed plant height, the increase in plant height in SRIBI was found to be statistically significant in comparisons with both SRI and CFR (Table 5). Here, AMD denotes the absolute mean difference, and its value should be higher than the critical Q value for the mean differences between two specific treatments to be deemed significant.

Number of Tillers

The number of tillers was found to be significantly higher in SRI-based management ($p \leq 0.05$; $p = 0.0001$) (Figure 3, Table 6). This is in line with the comparisons between SRI and flooded rice systems done in previous studies [27,53]. The higher number of tillers in

the SRI method has been attributed to the synergetic development of roots and tillers [54]. Tillering ability of rice has been directly associated with the rice grain yield, as the panicle number per hill is directly proportional to the total number of tillers, irrespective of whether the tillers are productive or unproductive [55]. In 2019, the mean number of tillers in SRIBI was found to be 26, while it was 9.5 for conventional flooded rice (CFR), visualized in Figure 3A comparison of the number of tillers in SRI, SRI with intercropping, and CFR, as observed in the field experiments in 2020, is also shown. For the 2020 data, a comparison of the means of all the three treatments revealed that the increase in the tiller number in the case of SRIBI was more significant than that observed in case of SRI treatment (Table 6).

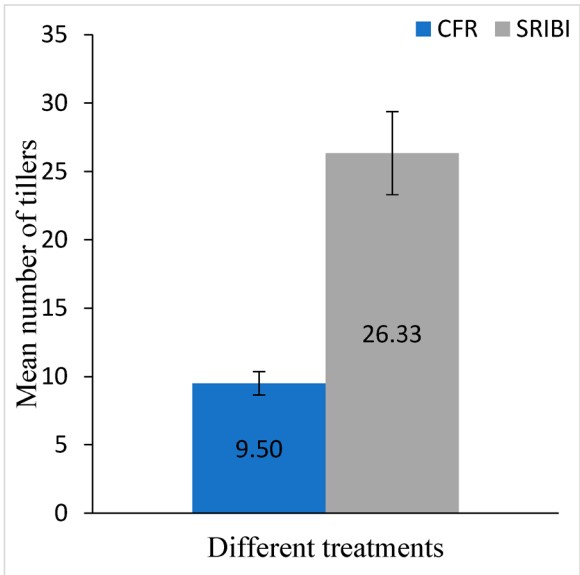
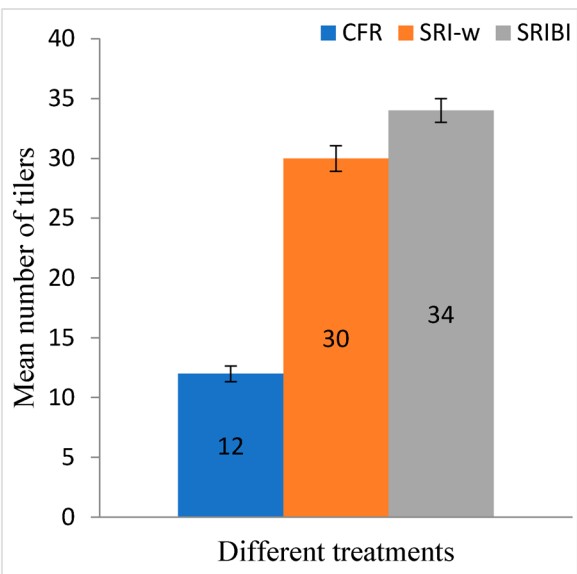

**Figure 3.** Mean number of tillers in the intercropping treatment and the conventional rice treatment (**Left**: 2019; **Right**: 2020).

**Table 6.** Parameters from ANOVA statistical analysis for tiller number comparison.

| Tiller Number | 2019 SRI-w vs. SRIBI | | | | 2020 CFR vs. SRI-w vs. SRIBI | | | |
|---|---|---|---|---|---|---|---|---|
| | *p*-value | F-value | $F_{critical}$ | DF | *p*-value | F-value | $F_{critical}$ | DF |
| | 0.0001 | 32.49 | 4.75 | (1,12) | 0.0001 | 160 | 3.56 | (2,18) |
| | | | | | Post hoc analysis | | | |
| | | | | | $Q_{critical}$ | $AMD_{CFR/SRI}$ | $AMD_{SRI/SRIBI}$ | $AMD_{CFR/SRIBI}$ |
| | | | | | 8.12 | 17.86 | 4.14 | 22 |

Panicle Length

Notwithstanding a higher number of tillers and, by extension, a higher number of panicles [55], in order to quantify the effects of intercropping on the yield, a comparison of the panicle length was done for all three treatments, flooded rice, SRI-w, and SRI with beans intercropping (SRIBI) (Figure 4). The difference among the three treatments was found to be statistically insignificant ($p > 0.05$; $p = 0.065$), indicating that the yield potential of rice was not negatively affected by either SRI or intercropping (Table 7). However, since the number of tillers and panicles in SRI-based management was multiple times higher, it was expected to translate to higher yields, as reported in other studies [55].

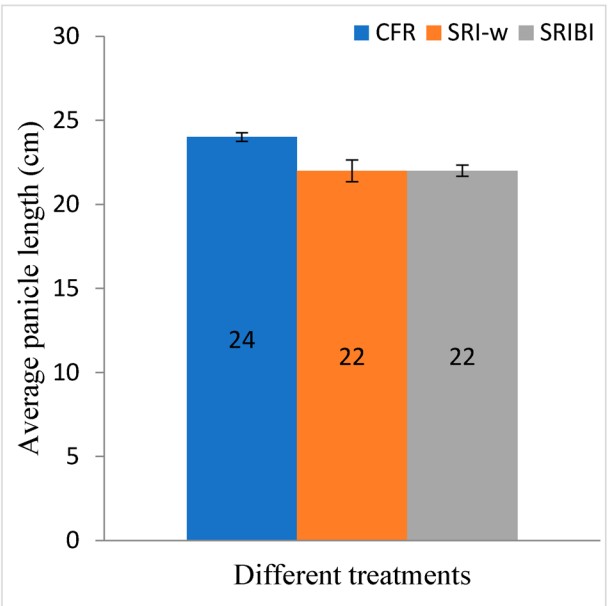 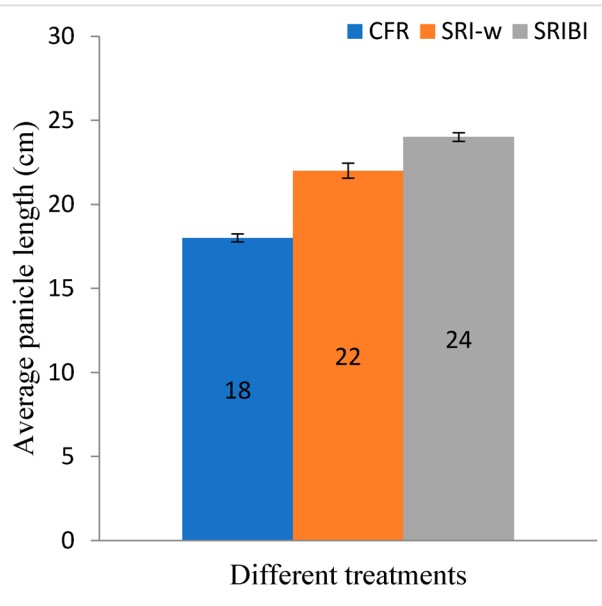

**Figure 4.** Averagepanicle length of the different treatments as observed in field experiments (**Left:** 2019; **Right**: 2020).

**Table 7.** Parameters from ANOVA statistical analysis for panicle length comparison.

| Panicle Length | 2019 SRI-w vs. SRIBI | | | | 2020 CFR vs. SRI-w vs. SRIBI | | | |
|---|---|---|---|---|---|---|---|---|
| | $p$-value | F-value | $F_{critical}$ | DF | $p$-value | F-value | $F_{critical}$ | DF |
| | 0.0649 | 3.12 | 3.35 | (2,7) | 0.0001 | 85 | 3.56 | (2,18) |
| | | | | | Post hoc analysis | | | |
| | | | | | $Q_{critical}$ | $AMD_{CFR/SRI}$ | $AMD_{SRI/SRIBI}$ | $AMD_{CFR/SRIBI}$ |
| | | | | | 0.89 | 3.93 | 2 | 5.93 |

There was a significant increase in this yield parameter, observed in the field experiments in 2020. In the case of panicle length as well, the increase in the case of SRIBI was found to statistically more significant (with a higher absolute mean difference) as compared to the increase in the case of SRI (Table 7). The difference of the means between SRIBI and SRI was also found to be significant.

Spikelet Number Per Panicle

Intercropping was found to have a positive effect on another yield parameter that was observed in the studies, the spikelet number per panicle (SNPP). The SNPP was found to be significantly higher in the case of rice under intercropping ($p \leq 0.05$; $p = 0.0001$). The average SNPP of rice with intercropping and without intercropping was observed as 142 and 95 in 2019 and 161 and 140 in 2020, respectively (Figure 5). When compared with the CFR treatment, the statistical significance of the SNPP in the case of SRIBI was observed to be higher as compared to that in SRI, based on the Tukey–Kramer post hoc test (Table 8). Additionally, the rice plots under the intercropping regimen were observed to be harvest-ready earlier than the plots without intercropping. In this regard, it is pertinent to report the chlorophyll content of the rice plant leaves that was measured under controlled conditions in greenhouse experiments in the next section. The SNPP has been associated with the soil microbial composition [56], indicating that intercropping might have modified the soil microbial composition, leading to an increase in the SNPP. Changes in soil bacterial communities favourable to rice yield have also been confirmed with other changes in the rice cropping system, such as double-season rice cropping [57].

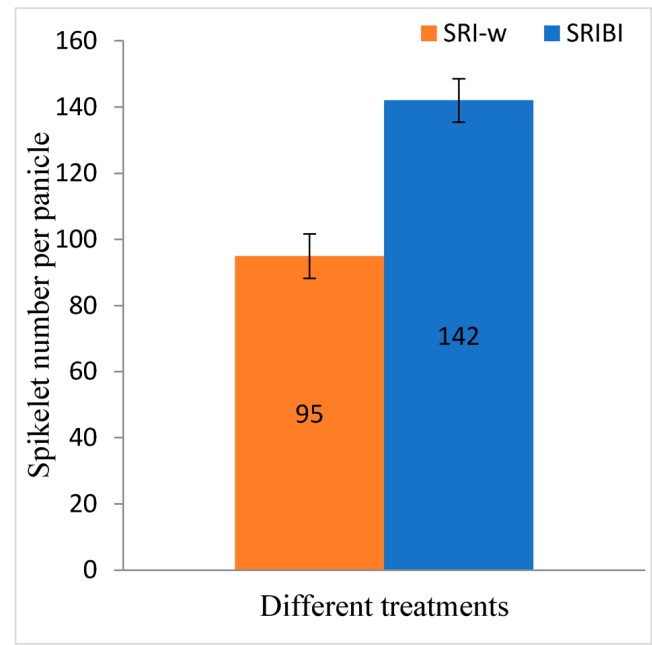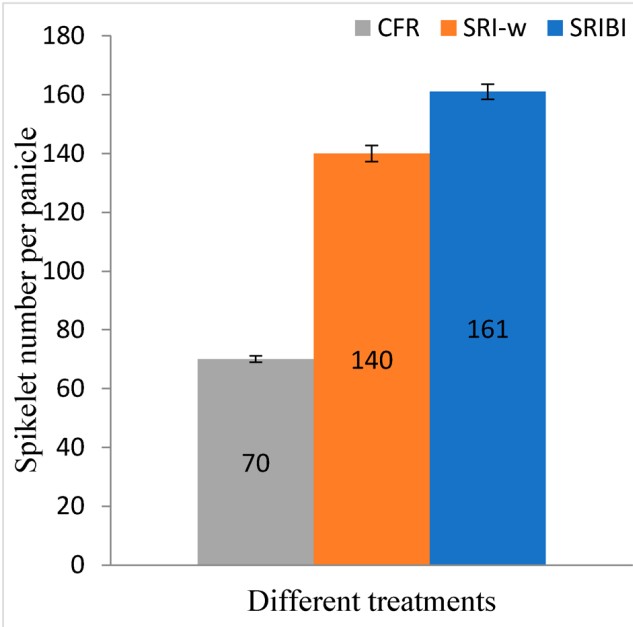

**Figure 5.** The spikelet number per panicle (SNPP) observed in the field experiments (**Left**: 2019; **Right**: 2020).

**Table 8.** Parameters from ANOVA statistical analysis for spikelet number (SNPP) comparison.

| SNPP | 2019 SRI-w vs. SRIBI | | | | 2020 CFR vs. SRI-w vs. SRIBI | | | |
|---|---|---|---|---|---|---|---|---|
| | *p*-value | F-value | $F_{critical}$ | DF | *p*-value | F-value | $F_{critical}$ | DF |
| | 0.0001 | 25.31 | 4.49 | (1,16) | 0.0001 | 329.80 | 3.40 | (2,24) |
| | | | | | Post hoc analysis | | | |
| | | | | | $Q_{critical}$ | $AMD_{CFR/SRI}$ | $AMD_{SRI/SRIBI}$ | $AMD_{CFR/SRIBI}$ |
| | | | | | 70.46 | 70.55 | 18.11 | 88.66 |

Filled Grains Per Panicle

The number of filled grains per panicle was counted for the three different treatments, and intercropping was found to have a positive effect on this parameter, as can be seen in Figure 6. The differences in this parameter are statistically significant in both SRI and SRIBI, as compared to CFR. However, the level of significance was higher in case of SRIBI, as can be seen from the AMD (absolute mean difference) (Table 9).

**Table 9.** Parameters from ANOVA statistical analysis for filled grains per panicle (FGPP) comparison.

| Grains Per Panicle | *CFR* vs. *SRI-w* vs. *SRIBI* | | | |
|---|---|---|---|---|
| | *p*-value | F-value | $F_{critical}$ | DF |
| | 0.0001 | 383 | 3.47 | (2,21) |
| | Post hoc analysis | | | |
| | $Q_{critical}$ | $AMD_{CFR/SRI}$ | $AMD_{SRI/SRIBI}$ | $AMD_{CFR/SRIBI}$ |
| | 65.43 | 73.5 | 21.625 | 95.125 |

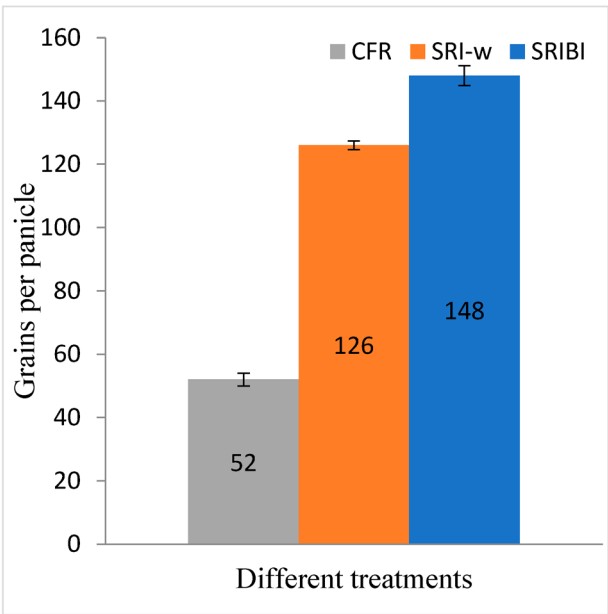

**Figure 6.** Grains per panicle observed in the different treatments (2020).

### 3.1.3. Rice Yield and Fodder Yield

Rice Yield

Based on the improvement in the plant growth parameters discussed in the previous sections, a higher grain yield was expected under SRIBI management. This was evident from the yields reported from the farmers' fields. The total yield measured in the field experiments of the year 2019 was higher under the SRIBI regime than the conventional flooded rice by 15–20%, and in 2020, it was found to be 33% higher under the SRI intercropping regime as compared to the conventional flooded rice cultivation (Figure 7).

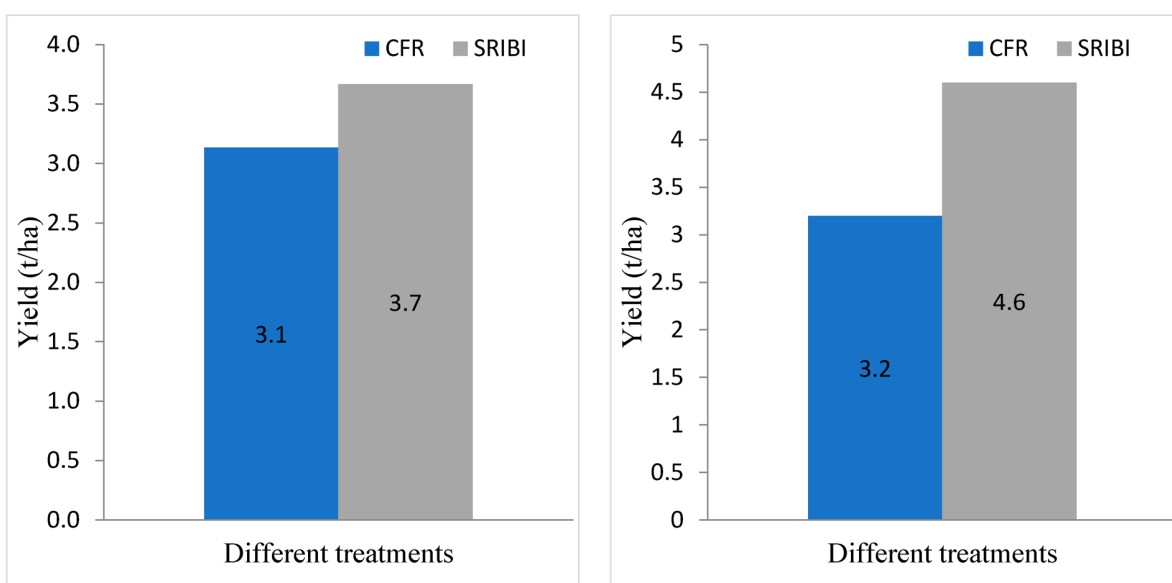

**Figure 7.** Yield observed in the different treatments (**Left**: 2019; **Right**: 2020).

Fodder Units

Rice husk is used either as fodder or as a filling material for horticultural practices and hence, has a significant economic value for the farmers. In 2019, the number of fodder units in CFR was found to be 90, while in case of SRI with intercropping, it was found

to be 140. Similarly, in 2020, the number of fodder units under intercropping SRI was found to be 160 (Figure 8). Hence, the trend in the case of rice husk biomass was similar to the grain yield. This can be attributed to the higher number of tillers in the case of SRI management practices.

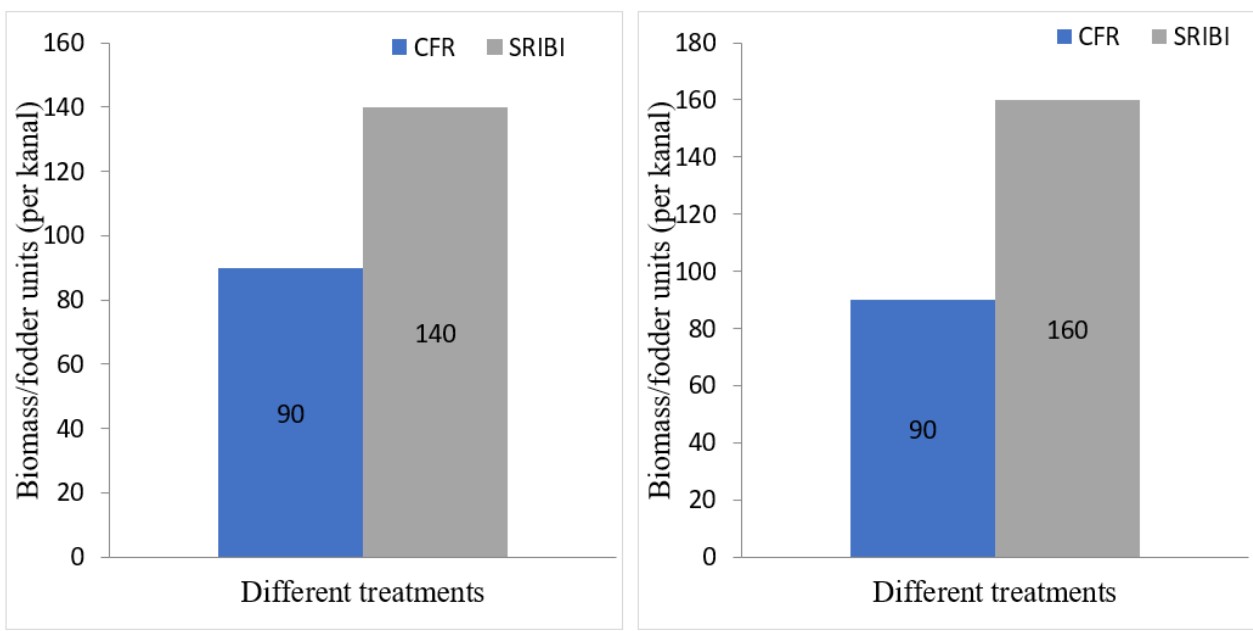

**Figure 8.** Number of fodder units observed in the different treatments (**Left**: 2019; **Right**: 2020).

A marked improvement was observed in different plant growth and yield parameters due to SRI and intercropping in comparison to the conventional flooded rice cultivation method. This indicates the potential of these methodologies in improving the yield of rice farming systems, which can result in improvements in the food security scenario and the socio-economic condition of rice farmers.

### 3.1.4. Economic Balance

Work-Hours

With respect to the other main criticism of the SRI methodology, which is that it leads to an increase in work-hour requirements due to the need of manually operated weeding, the intercropping regimen was beneficial. It was observed that for 1 kanal (0.05 ha) of land, 1 h of time was required for one round of weeding, which translates to 20 extra work-hours per hectare of land, per weeding. With the standard of 4 weedings per cycle of rice production in SRI, this translates to 80 work-hours per hectare. Assuming that intercropping is done after the first weeding, as was done in the current study, this can lead to savings of 75% of the extra work-hours required by SRI, which is 60 work-hours per hectare. This is a conservative estimate, given that higher work-hour requirements for weeding, of up to 160 work-hours per hectare per cycle of rice production under SRI-based rice farming techniques, have been reported. Thus, the work-hour savings could be even higher, depending on the stage of adoption of the SRI method [32].

Economic Balance Sheet

The economic balance of the new rice farming system with beans as the intercrop, as seen from the field experiments, is presented in Table 10. The increase in the net earnings of farmers was observed to be 57%. This was accompanied by an increase of 41% in the input costs with a corresponding increase of 51% in the output benefits. The cost of compost constituted 63% of the input costs, and this points to the potential of decreasing the inputs

costs even further through local production of compost (Table 10). In this case, the net earnings would be more than double compared to the conventional rice farming system.

**Table 10.** Economic balance sheet of the innovations implemented during the course of research.

| | Input | Quantity (per ha) | Cost (INR) | Output | Quantity (per ha) | Benefit (INR) | Earnings (INR) | Increase (%) |
|---|---|---|---|---|---|---|---|---|
| CFR | Seeds | 400 kg | 6000 | Rice | 5600 kg | 84,000 | | |
| | Manure | 0.33 trolley | 20,000 | Fodder (Rice Straw) | 1800 units | 72,000 | | |
| | Fertilizer | 300 kg | 6666 | | | | | |
| | Pesticides | n.d. | 4000 | | | | | |
| | Labour for weeding | 80 work hours | 5000 | | | | | |
| | Labour for irrigation | 200 work hours | 12,500 | | | | | |
| | **Total cost** | | **54,166** | **Total benefit** | | **156,000** | **101,834** | |
| SRIBI | Seeds | 40 kg | 600 | Rice | 7200 kg | 108,000 | | |
| | Manure | 0.33 trolley | 20,000 | Fodder (Rice Straw) | 3200 units | 128,000 | | |
| | Compost | 6000 kg | 48,000 | | | | | |
| | Intercrop Seeds | 2 kg | 400 | | | | | |
| | Labour for weeding | 10 work hours | 660 | | | | | |
| | Labour for watering | 100 work hours | 6260 | | | | | |
| | **Total cost** | | **75,920** | **Total benefit** | | **236,000** | **160,080** | **57.20** |

Disease Incidence in Mushkibudij (Mushk Budji) Rice Landrace

The study experiments were conducted with a local heritage aromatic rice landrace known as Mushkibudij. This landrace of rice was revived by the local agricultural university and has been reported to be susceptible to rice blast that is the most widely occurring disease in rice, which has led to huge losses in yield, of up to 70% [58–60]. This leads to huge losses for the farmers while also increasing the input costs in the form of insecticide sprays. However, in the case of current experiments, the incidence of rice blast was not observed in any of the plots grown under SRI and SRIBI. The wider spacing between rice plants under SRI management system could be one of the factors behind this decreased incidence of disease [61,62]. This could form a basis for further research in this direction.

*3.2. Pot Experiments*

Pot experiments were conducted to establish the proof of concept about the positive effect of intercropping under the system of rice intensification on rice crop. The following parameters were measured.

3.2.1. Nutrient Uptake

The rice plants were analysed for nutrient uptake in the pot experiments performed at the greenhouse level. The content of the three essential nutrients, nitrogen (Figure 9), potassium (Figure 10), and phosphorous (Figure 11), was found to be higher in the case of SRI with intercropping as compared to the treatment without intercropping.

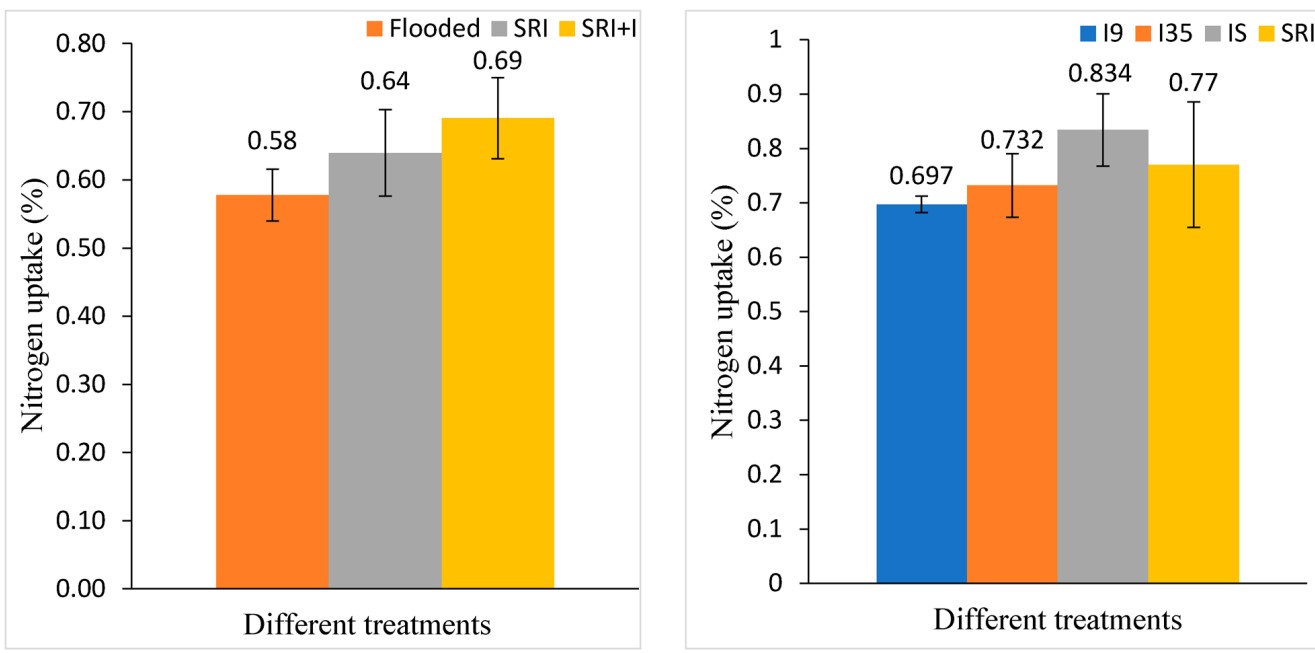

**Figure 9.** Nitrogen uptake as measured in the different treatments. Conventional flooded rice (CFR), SRI, and SRI with intercropping (SRI+I or SRIBI) in 2017 (**Left**). SRI and three different intercropping configurations in 2018 (**Right**).

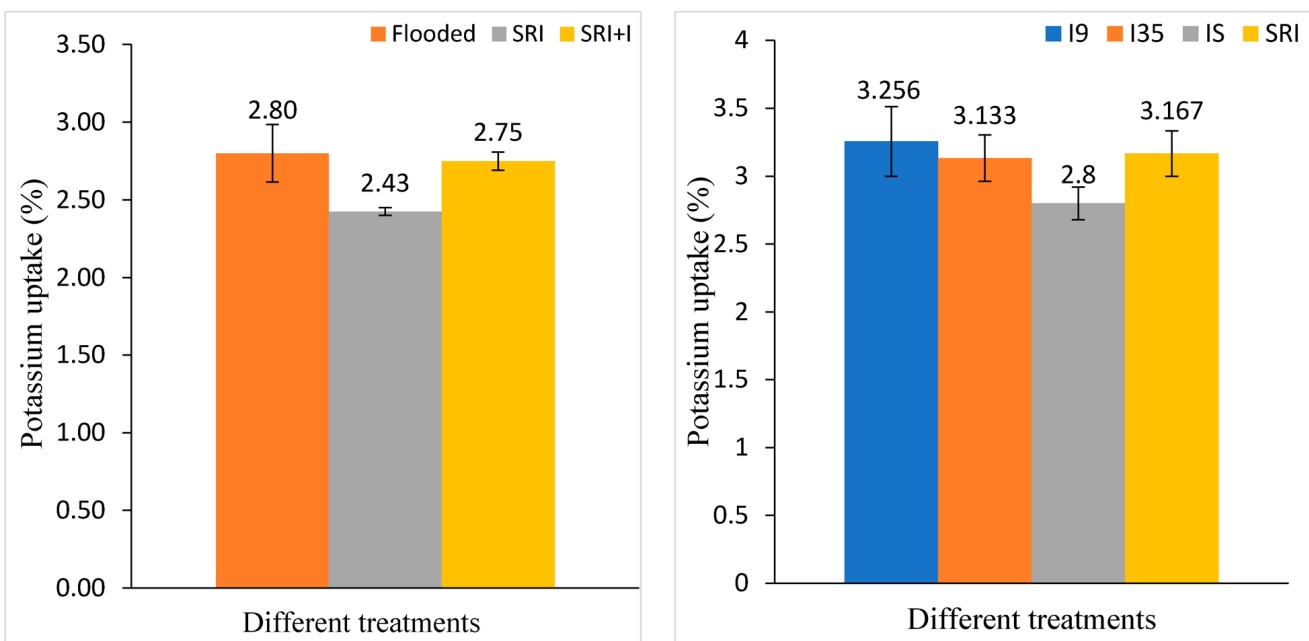

**Figure 10.** Potassium uptake as measured in the different treatments. Conventional flooded rice (CFR), SRI, and SRI with intercropping (SRI+I or SRIBI) in 2017 (**Left**). SRI and three different intercropping configurations in 2018 (**Right**).

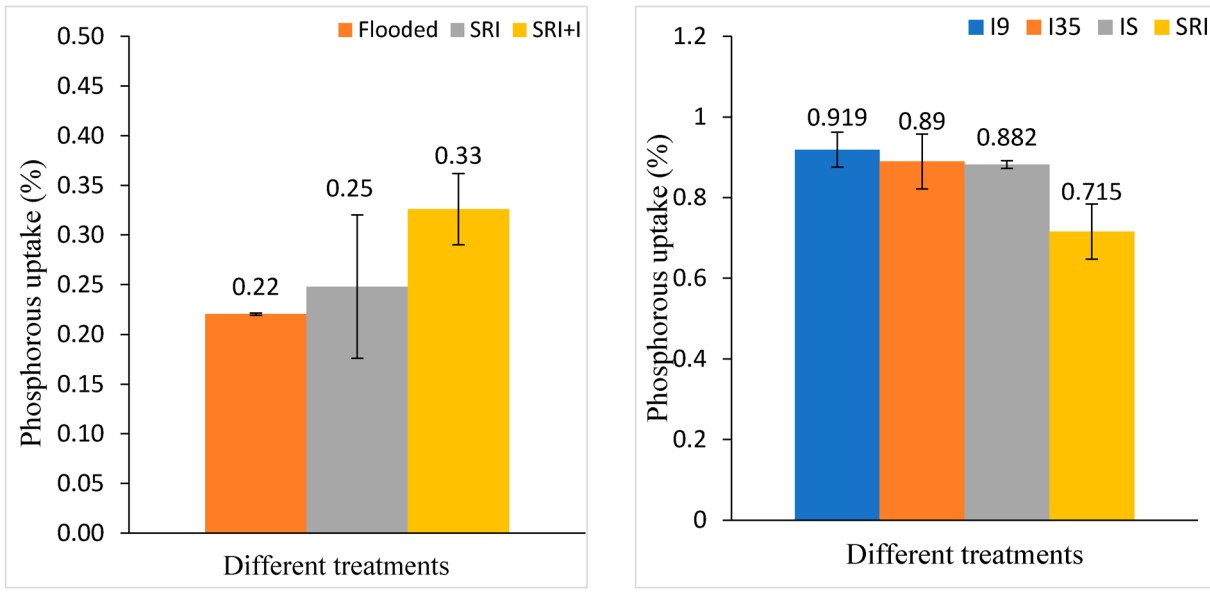

**Figure 11.** Phosphorous uptake as measured in the different treatments. Conventional flooded rice (CFR), SRI, and SRI with intercropping (SRI+I or SRIBI) in 2017 (**Left**). SRI and three different intercropping configurations in 2018 (**Right**).

As a general trend, the uptake of essential nutrients, nitrogen, phosphorous, and potassium, was found to be higher with intercropping as compared to the control SRI treatment.

### 3.2.2. Chlorophyll Content

A significant difference was observed in the Chlorophyll-a and Chlorophyll-b content of the rice plants grown under SRI and SRIBI ($p \leq 0.05$). A higher chlorophyll content in SRI as compared to the conventional flooded rice (CFR) has also been reported in previous studies [63]. While the mean value of Chlorophyll-a was 9.3 for CFR and SRI, it was 13.2 for SRIBI (SRI+I), which was significantly higher ($p = 0.05$) than that of SRI (9.2) (Figure 12). On the other hand, the mean value of Chlorophyll-b was 3.0 for CFR, while it was 4.1 for SRIBI, which was significantly higher ($p = 0.035$) than that of SRI (2.9) (Figure 13) (Table 11).

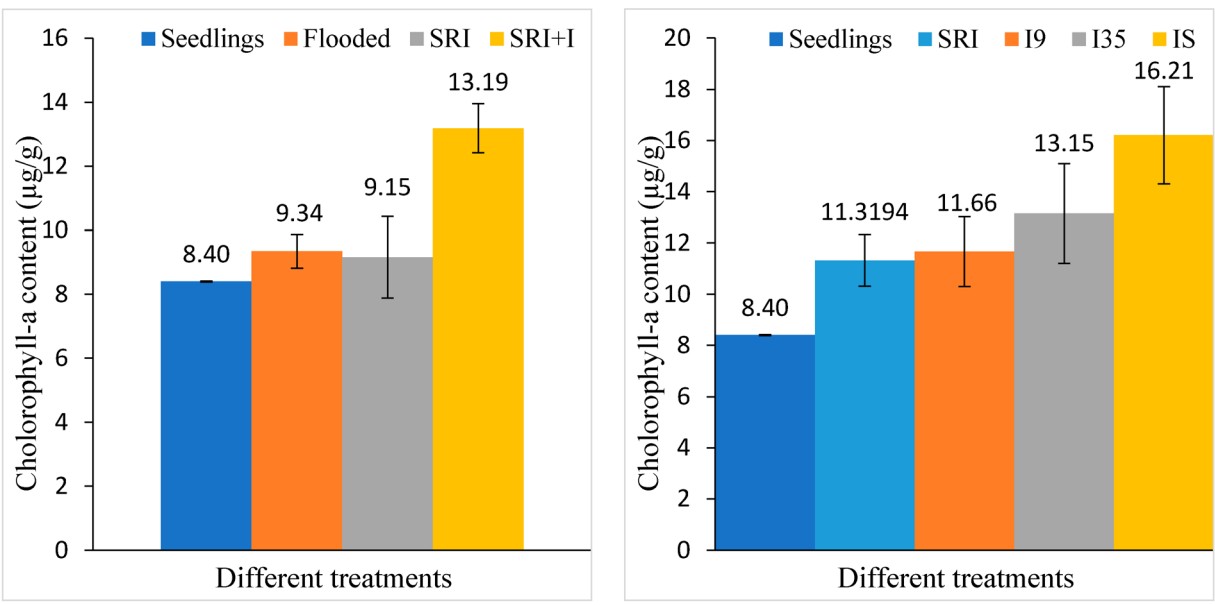

**Figure 12.** Comparison of chlorophyll-a content between the different treatments (**Left**: 2017; **Right**: 2018).

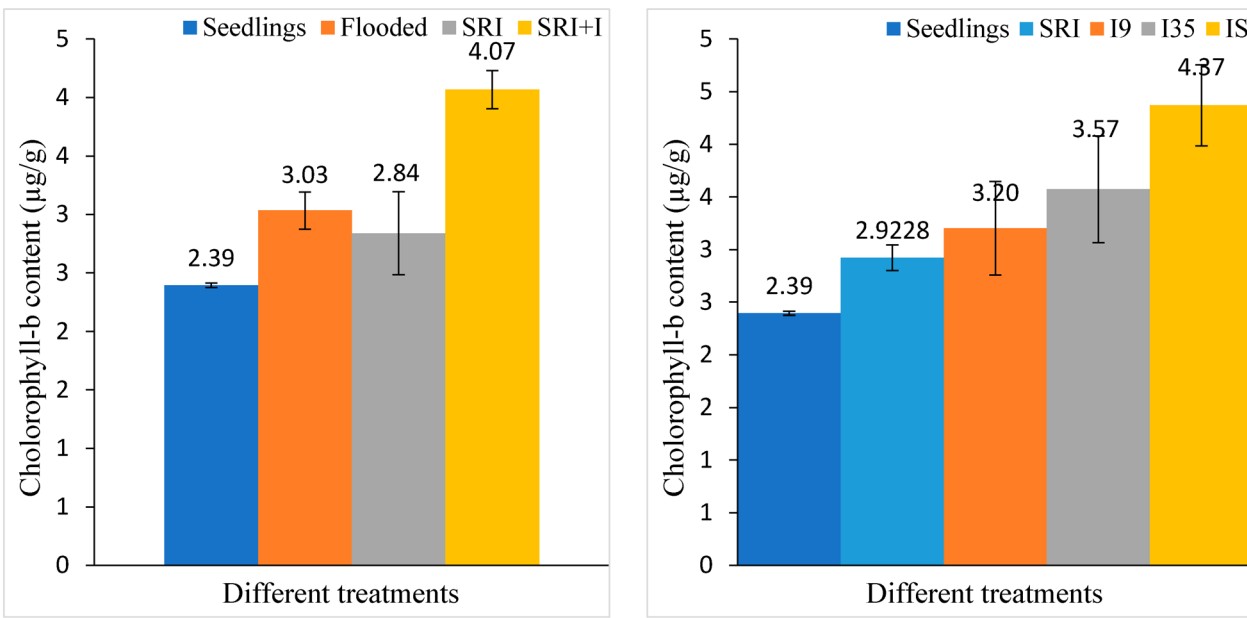

**Figure 13.** Comparison of chlorophyll-b content between the different treatments (**Left**: 2017; **Right**: 2018).

**Table 11.** Parameters from ANOVA statistical analysis for chlorophyll content comparison.

| *Chlorophyll Content* | **Chlorophyll-a SRI vs. SRIBI** | | | | **Chlorophyll-b SRI vs. SRIBI** | | | |
|---|---|---|---|---|---|---|---|---|
| | *p*-value | F-value | $F_{critical}$ | DF | *p*-value | F-value | $F_{critical}$ | DF |
| | 0.05 | 7.39 | 7.70 | (1,4) | 0.035 | 9.87 | 7.71 | (1,4) |

Chlorophyll content was also measured in three different intercropping treatments, I9 (intercropping at 9 days after transplantation), I35 (intercropping at 35 days after transplantation), and IS (strip intercropping). The chlorophyll content in the three different intercropping treatments (I9, I35, IS) was not found to be statistically significantly different from each other ($p > 0.05$; $p = 0.1153$, $p = 0.1524$) (Figures 12 and 13). This indicates that while intercropping had an effect on the chlorophyll content of the rice plant, the time and spatial differences in intercropping did not have a significant effect.

A progressive decrease in the chlorophyll content in rice has been associated with the ripening of the rice plants [53,56]. This decrease in chlorophyll content was associated with senescence of the rice plants, which has been found to occur earlier in flooded rice than in the SRI-based techniques [53,56]. A higher chlorophyll content in the leaves has been linked to a higher photosynthesis rate and root activity [63].

### 3.2.3. Tiller Number

In pot experiments, the number of tillers was found to be the highest in the case of SRI with intercropping, followed by SRI. The conventional flooded rice (CFR) treatment had the lowest number of tillers (Figure 14).

The parameters measured in the pot experiments described above provided the required proof of concept to perform the intercropping experiments at the field level. The results of the field experiments conclusively indicated the potential of the intercropping innovation in making rice farming a socially and ecologically sustainable agricultural production system.

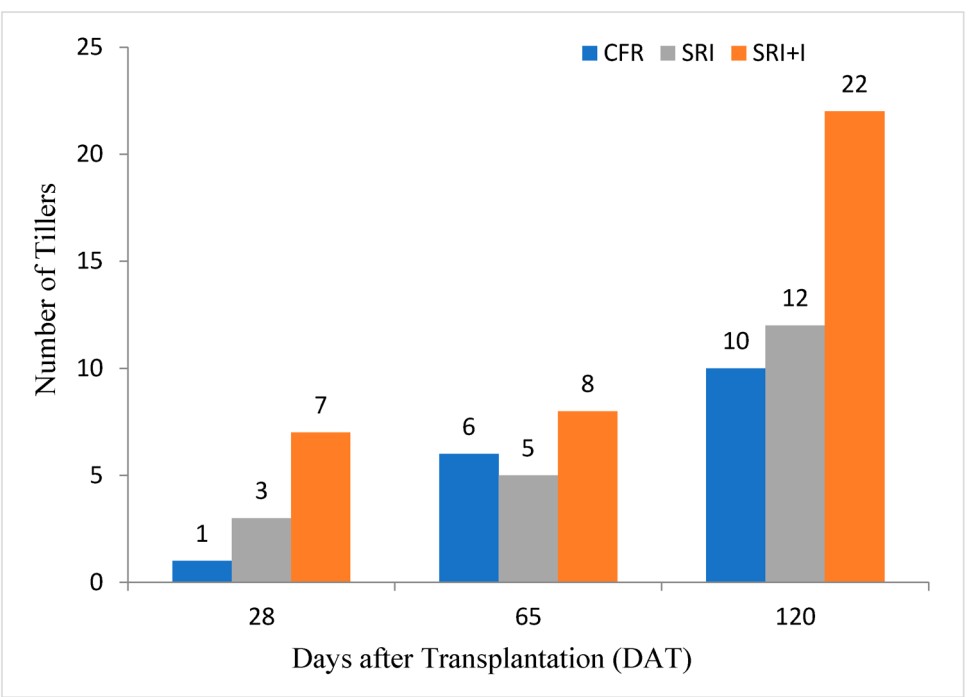

**Figure 14.** Comparison of the highest number of tillers observed in different phonological stages in the three different treatments: CFR, conventional flooded rice; SRI; SRI+I (SRIBI).

## 4. Conclusions

The results of experiments carried out at the lab scale in greenhouse (pot experiments) provided the proof of concept about the value addition that intercropping could provide in rice cultivation under the System of Rice Intensification (SRI). The results from the field experiments in 2019 and 2020 concurred with the findings of the pot experiments carried out from 2017 to 2018. The findings show that intercropping beans in the inter-row space in rice planted under the SRI management system provides value addition to the farmers, mainly visible in the form of reduced work hours and increased yields.

This study provided further clarity that SRI-based rice cultivation can be more productive and beneficial for farmers, while at the same time presenting SRIBI as an improved agroecological developmental stage of SRI. Intercropping serves to address two oft-cited criticisms of SRI, weed infestation and increased labour requirement, which have been linked to its dis-adoption. The improvements resulting from intercropping observed in this study include a clear reduction in weed infestation, higher nutrient uptake, improved plant growth characteristics, and better yield parameters. These improvements have the potential to control the overuse of pesticides in rice farming, which has been linked to human and planetary health concerns.

These improvements highlight the potential of such agroecological interventions to improve the socio-economic condition of rice farmers significantly, especially smallholder and subsistence farmers who otherwise depend on various agrochemical inputs. Intercropping legumes with rice can also lead to a diversification of farmers' incomes as well as their diets. This can have positive implications on the livelihood and food security scenario given that rice is a staple for more than half of the world's population.

**Author Contributions:** Conceptualization, T.M.S. and R.O.; methodology, T.M.S.; software, S.T.; validation, T.M.S., R.O. and M.A.B.; formal analysis, T.M.S.; investigation, T.M.S.; resources, T.M.S. and R.O.; data curation, S.T. and M.A.B.; writing—original draft preparation, T.M.S.; writing—review and editing, S.T.; visualization, T.M.S.; supervision, R.O. and M.A.B.; project administration, T.M.S. All authors have read and agreed to the published version of the manuscript.

**Funding:** This research received no external funding.

**Institutional Review Board Statement:** Not applicable.

**Informed Consent Statement:** Not applicable.

**Data Availability Statement:** Data are contained within the article. Further details about this study is available online here: https://doi.org/10.15480/882.3286 (Last accessed on: 19 May 2021).

**Acknowledgments:** We acknowledge support for Open Access publishing through APC funding by GFEU e.V. at TUHH. We acknowledge the financial support provided to the first author by the State of Hamburg (Free and Hanseatic City of Hamburg) through the HmbNFG (Hamburg State Graduate Funding Program) doctoral scholarship. We acknowledge the support provided to the first author by CCAFS-CGIAR through the CLIFF (Climate Food and Farming) fellowship. We acknowledge the feedback provided by Norman Uphoff during this research and the pioneering role played by SRI-RICE (SRI International Network and Resources Centre) in promoting socially, economically, and ecologically sustainable farming systems worldwide.

**Conflicts of Interest:** The authors declare no conflict of interest.

## Appendix A

**Table A1.** Different Statistical Data about the Different Measured Parameters.

| | Mean | | | Standard Deviation | | | Standard Error | | |
|---|---|---|---|---|---|---|---|---|---|
| | CFR | SRI | SRIBI | CFR | SRI | SRIBI | CFR | SRI | SRIBI |
| Plant height *(Field 2020)* | 101 | 104 | 109 | 1.14 | 1.95 | 1.82 | 0.51 | 0.87 | 0.81 |
| Plant height *(Field 2019)* | 125 | 119 | 128 | 4.88 | 4.59 | 9.56 | 1.47 | 1.38 | 3.38 |
| Panicle length *(Field 2020)* | 18 | 22 | 24 | 0.65 | 1.17 | 0.67 | 0.24 | 0.44 | 0.25 |
| Panicle length *(Field 2019)* | 24 | 22 | 22 | 0.81 | 2.03 | 1.05 | 0.25 | 0.64 | 0.34 |
| Tiller Number *(Field 2020)* | 12 | 30 | 34 | 1.72 | 2.82 | 2.64 | 0.65 | 1.07 | 0.99 |
| Tiller Number *(Field 2019)* | 9.5 | n.a. | 26.33 | 2.15 | n.a. | 7.45 | 0.81 | n.a. | 2.82 |
| SNPP *(Field 2020)* | 70 | 140 | 161 | 3.27 | 8.13 | 7.78 | 1.09 | 2.71 | 2.59 |
| SNPP *(Field 2019)* | n.a. | 95 | 142 | n.a. | 20.12 | 19.79 | n.a. | 6.70 | 6.59 |
| FGPP *(Field 2020)* | 52 | 126 | 148 | 6.04 | 4.10 | 9.55 | 2.01 | 1.36 | 3.18 |
| Chlorophyll-A | 9.3 | 9.2 | 13.2 | 0.9 | 2.2 | 1.3 | 0.5 | 1.3 | 0.8 |
| Chlorophyll-B | 3.0 | 2.8 | 4.1 | 0.3 | 0.6 | 0.3 | 0.2 | 0.4 | 0.2 |
| N-content | 0.58 | 0.64 | 0.69 | 0.10 | 0.10 | 0.08 | 0.06 | 0.04 | 0.03 |
| P-content | 0.22 | 0.25 | 0.33 | 0.01 | 0.04 | 0.05 | 0.003 | 0.02 | 0.02 |
| K-content | 2.83 | 2.43 | 2.75 | 0.15 | 0.08 | 0.07 | 0.09 | 0.04 | 0.03 |

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
