# Peer review of "Intercropping in Rice Farming under the System of Rice Intensification—An Agroecological Strategy for Weed Control, Better Yield, Increased Returns, and Social–Ecological Sustainability"

_agronomy, doi:10.3390/agronomy11051010_

Round 1

Reviewer 1 Report

The manuscript submitted for review takes up very important issues in rice production and sustainable agriculture. It is right to take up the subject of the Authors' research, because the work is interesting and practically useful. The research undertaken is in line with the current global trends of seeking solutions to save water, fertilizers and energy. The work fits in with the research subject of Agronomy.

The manuscript after completion and corrections deserves publication.

Comments and suggestions

Line 80: should be- 40% [11] and everywhere throughout the text, spaces between the text and the reference number, e.g. line 82: health [25–28].

Line 89: should be- specifically in organic farming systems[33–35].

Line 104: it would be good to describe briefly principles of system of rice intensification (SRI), because there is little about it in the text of the manuscript.

Line 106-107: should be growing leguminous crop –beans, or only leguminous crop, or only beans.

Line 117-118: what was the size (capacity) of the pots?

Line 120-122: what substrate (soil)? Was used in the pot experiment or what nutrient medium?

Line 132-133: in what period was carried out, months?

Line 140-141: necessary supplement - meteorological data (temperature, rainfall, number of sunny days). Chemical analysis of the soil on which the experiment was carried out, type of soil.

Line 150: should be: standard application: 300 kg (150 N +75 P2O5+75 K2O) per ha.

Line 168-172: chlorophyll a and b contents was described in Mackinney (1941) and Arnon (1949), and it is so commonly quoted, mainly as Arnon (1949).

Results and Discussion

Figure 1. There is no sign in the current figure which chart is for 2019, whether the one on the left or on the right. Likewise, all other figures in the manuscript. A location of the data might be more legible on a single figure with 2019 and 2020 year.

There should be an explanation of abbreviations under the figures, tables and photos, e.g. SRI- System of Rice Intensification

Some figures can be omitted in the publication, e.g. figure 13. And figure 12 summarize the content of total chlorophyll (a + b).

References: there are technical errors, sometimes no pages or online availability e.g.

Line 597: should be: Washington

Line 598: is available on line, pages, accessed on…?

Line 627: FAO

Line 628: is available on line, pages, accessed on…?

Line 630: pages?

Line 684: should be 2018 (position 48 - Doni et al.).

Author Response

  • The manuscript submitted for review takes up very important issues in rice production and sustainable agriculture. It is right to take up the subject of the Authors' research, because the work is interesting and practically useful. The research undertaken is in line with the current global trends of seeking solutions to save water, fertilizers and energy. The work fits in with the research subject of Agronomy. The manuscript after completion and corrections deserves publication. Thank you for your detailed feedback and suggestions for improving the manuscript.
  • Comments and suggestions
  • Line 80: should be- 40% [11] and everywhere throughout the text, spaces between the text and the reference number, e.g. line 82: health [25–28]. DONE
  • Line 89: should be- specifically in organic farming systems[33–35]. DONE
  • Line 104: it would be good to describe briefly principles of system of rice intensification (SRI), because there is little about it in the text of the manuscript. ADDED AT 78-80
  • Line 106-107: should be growing leguminous crop –beans, or only leguminous crop, or only beans. DONE
  • Line 117-118: what was the size (capacity) of the pots? ADDED at Lines 124 and 133-134
  • Line 120-122: what substrate (soil)? Was used in the pot experiment or what nutrient medium? Details added at Lines 122-124
  • Line 132-133: in what period was carried out, months? Added at Line 139
  • Line 140-141: necessary supplement - meteorological data (temperature, rainfall, number of sunny days). Added at Lines 146-153 (Data on the number of sunny days is not available).
  • Chemical analysis of the soil on which the experiment was carried out, type of soil. DONE (146-153) 
  • Line 150: should be: standard application: 300 kg (150 N +75 P2O5+75 K2O) per ha. DONE
  • Line 168-172: chlorophyll a and b contents was described in Mackinney (1941) and Arnon (1949), and it is so commonly quoted, mainly as Arnon (1949). ADDED at Line 180
  • Results and Discussion
  • Figure 1. There is no sign in the current figure which chart is for 2019, whether the one on the left or on the right. Likewise, all other figures in the manuscript. A location of the data might be more legible on a single figure with 2019 and 2020 year. DONE.
  • The reference to the years 2019 and 2020 has been added to the figure captions. There should be an explanation of abbreviations under the figures, tables and photos, e.g. SRI- System of Rice Intensification. The explanation of the abbreviations has been mentioned in the text part in Abstract, Introduction, Materials and Methods, and Conclusions multiple times. The authors feel it would become unnecessarily repetitive if the expanded form of the abbreviations were to be added under every figure, table, and photo. For ease of following the results, an explanation of the abbreviations has been added under the figures where more clarity was needed. A note has been added at Line 220 as well.
  • Some figures can be omitted in the publication, e.g. figure 13. And figure 12 summarize the content of total chlorophyll (a + b). Thank you for the suggestion. However, the figures have been retained just for some extra clarity about the two constituents (a and b).
  • References: there are technical errors, sometimes no pages or online availability e.g. DONE The missing information has been added at the mentioned places and others.

Reviewer 2 Report

Abstract - No specific comments.

Introduction - No specific comments.

Materials and Methods – there are no data about weed species present in the experiment. These data were not subject of detailed analysis in Results, but it would be fine to have some info about the main weed species in this section.

Results and Discussion – The presentation of the results should be improved. There are too many tables with relatively long headings and only a few rows. I recommend to present the data in few but longer tables. Most of the figures contain standard deviations, but not the figures 7, 8, 14.

Conclusions – No specific comments.

Author Response

Thank you for your suggestions for improving the manuscript.

Materials and Methods – there are no data about weed species present in the experiment. These data were not subject of detailed analysis in Results, but it would be fine to have some info about the main weed species in this section. ADDED at Line 225-226

Results and Discussion 

There are too many tables with relatively long headings and only a few rows. I recommend to present the data in few but longer tables.

The authors opine that the close placing of the figures and the corresponding tables can give the reader a better idea about the significance of the results.

Most of the figures contain standard deviations, but not the figures 7, 8, 14.

The data in Figures 7 and 8, in contrast to the plant-specific data in the other figures, is a summative data taken at the plot level, for comparison of the conventional method and the intercropping method.

Figure 14 is a comparison of the highest Tiller Number overall observed in a particular treatment.

Reviewer 3 Report

Improving rice growing technology is very important in the today situation because SRI-based rice farming systems have been associated with a range of environmental benefits: reducing water and energy use and greenhouse gas emissions.

However, the experimental design needs to be revised. The „Seedlings“ treatment appears in Figures 12 and 13. The results of this treatment should also be described. You do not write about this treatment in the subsection 2.1. Experimental research design.

I made a few observations in the manuscript.

Author Response

Thank you for your positive feedback and suggestions for improving the manuscript.

However, the experimental design needs to be revised. The „Seedlings“ treatment appears in Figures 12 and 13. The results of this treatment should also be described. You do not write about this treatment in the subsection 2.1. Experimental research design.

The ‘Seedlings’ in Figures 12 and 13 refers to the chlorophyll content measured in the seedlings that were used in the treatments. It is mentioned in Line 182 in subsection 2.3.

I made a few observations in the manuscript.

Thank you for the comments in the manuscript. I have made the necessary changes and made additions where necessary.

Round 2

Reviewer 1 Report

The authors have now provided a revised version, the comments made in the first version of the work were included in the reviewed manuscript.

I believe the manuscript has been  sufficiently improved to warrant publication in Agronomy.